# HIGH-FIDELITY PRUNING FOR LARGE LANGUAGE MODELS

## ABSTRACT

Large Language Models (LLMs) have demonstrated exceptional performance across a wide range of tasks, yet their significant computational and memory requirements present major challenges for deployment. A common approach uses Taylor expansion on the loss function to estimate neuron importance. However, its reliance on one-hot cross entropy loss, a key limitation is that it narrowly assesses importance based only on the probability assigned to the single predicted next token, thereby ignoring the other potential predictions of the original model. An intuitive solution to address this is to employ self distillation criterion for importance evaluation. However, this approach introduces significant computational overhead by requiring a separate teacher model for supervision. To this end, we propose a simple but effective criterion, information entropy of the model's output distribution, to efficiently evaluate importance scores of neurons with Taylor pruning without requirement of additional teacher. Compared to plain cross entropy criterion, it provides a more holistic criterion for Taylor pruning to prune neurons with the least impact on the prediction of model in a global manner, thereby preserving the fidelity of the model's predictive capabilities. Experimental results on extensive zero-shot benchmarks demonstrate that our method consistently outperforms existing pruning methods across the LLaMA and Qwen series models. The source code and trained weights will be publicly available.

## 1 INTRODUCTION

Large Language Models have become foundational in modern artificial intelligence (OpenAI et al., 2024; Touvron et al., 2023; Bai et al., 2023; DeepSeek-AI et al., 2024). This success, however, is built upon models with billions of parameters, whose substantial computational and memory footprints present significant barriers to their widespread deployment, particularly in resource-constrained environments. To reduce computational and memory footprints of LLMs, we analyze the proper components for pruning from two aspects, the proportion of parameters and the operational risk of performance degradation. We find that Multi-Layer Perceptron (MLP) modules constitute the vast majority of parameters in modern LLMs, thus offering the most significant opportunity for parameter reduction (Kaplan et al., 2020). For instance, in the Llama2-7B model, MLP modules account for approximately 68.3% of the parameters of the whole model. Moreover, we observe that pruning attention heads is a coarse-grained operation, akin to removing entire functional units (Voita et al., 2019), which poses a high risk of catastrophic performance degradation. Therefore, in this paper, we focus on the pruning of MLP modules to facilitate a more stable trade-off between model capacity and efficiency.

To minimize performance loss after pruning, Taylor-based pruning methods (Molchanov et al., 2017; 2016; 2019) estimate parameter importance through its first-order influence on a given loss function using Taylor expansion and prune the parameters with the least importance. However, as shown in Figure 1 (a), the efficacy of these methods is fundamentally hampered by their reliance on one-hot cross entropy criterion, which measures how accurately the model predicts a single, ground-truth next token. This criterion is designed to assess importance based only on the single ground-truth next token, while ignoring all other potential predictions. Consequently, the pruning process guided by this criterion only minimizes the change of label-related prediction, failing to preserve the rich knowledge encoded across the model's full output distribution.

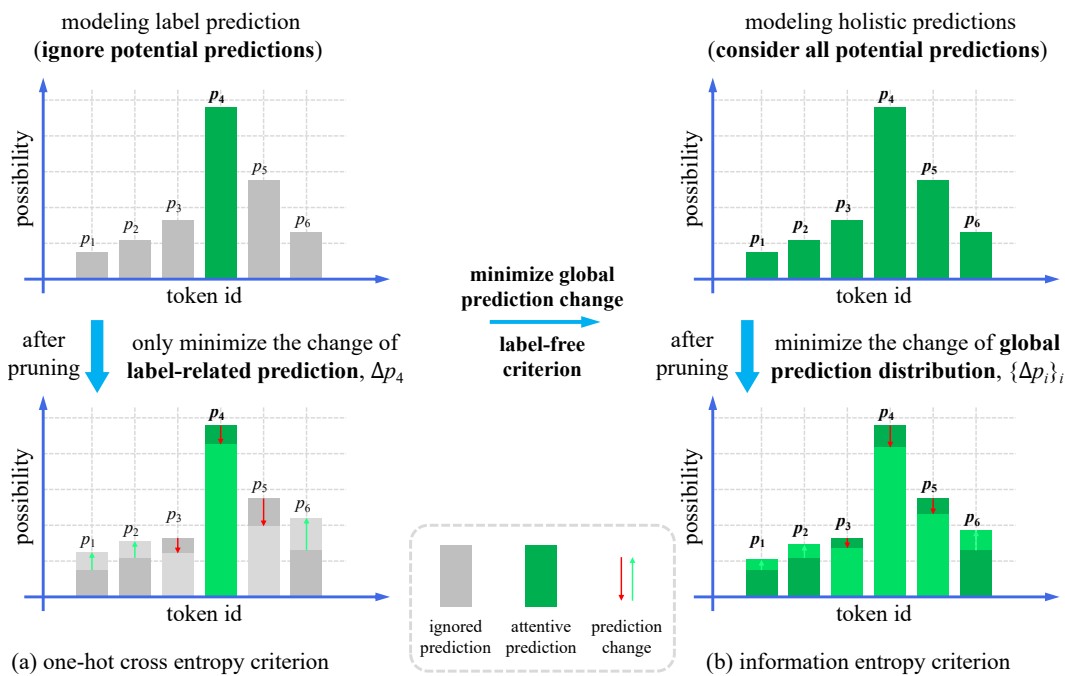

Figure 1: One-hot label loss criterion *vs* our proposed information entropy criterion. The cross entropy criterion (**left**) measures the model prediction by only label-related prediction. Based on this criterion, Taylor-based pruning only minimizes the change of label-related prediction after pruning. In contrast, our proposed information entropy (**right**) fully represent holistic predictions, thus minimizing the change of global prediction distribution for better performance preservation.

An intuitive solution is to employ self distillation criterion for importance evaluation as SDM-Prune (Zhu & Shen, 2025). However, this approach suffers from two significant drawbacks. It not only introduces significant computational overhead by requiring a separate teacher model for supervision, but it also suffers from a more critical defect, where the initial distillation loss is zero, leaving no gradient to guide the initial importance scoring.

To address the above issues, we propose a novel criterion for Taylor based neuron importance evaluation. Instead of conventional loss function, we introduce information entropy of the model's prediction distribution as the core criterion for Taylor based importance evaluation as Figure 1 (b). This shift in criterion offers two distinct advantages over existing methods. First, unlike the narrow focus of cross entropy on a single target token, our criterion can model holistic distribution of potential predictions. It provides a more faithful measure of neurons' contribution by considering all potential predictions, ensuring the pruning process aims to minimize the change of the global prediction distribution. Second, our method is conceptually simpler and more efficient than self distillation approaches like SDMPrune. It circumvents the need for a separate teacher model, eliminating the associated computational overhead and avoiding the critical flaw of a null initial gradient. Our complete process involves pruning neurons with the lowest entropy-based importance scores, followed by a brief fine-tuning phase to efficiently restore the model's performance.

Overall, our main contributions can be summarized as follows.

- We introduce a novel pruning criterion based on information entropy for Taylor-based pruning, which creates an elegant, efficient, and fundamentally label-free criterion for importance scoring.

- By modeling holistic predictions, our entropy-based criterion results in a more accurate importance assessment, which better preserves the model's intrinsic knowledge after pruning by minimizing the change of the global prediction distribution.

- Extensive zero-shot benchmarks demonstrate that our method consistently outperforms existing techniques on the LLaMA and Qwen series models. Especially for LLaMA2-7B,

with 20% parameters and FLOPs reduction, our pruned model not only recovers but even exceed the performance of the original dense model after a brief fine-tuning.

## 2 RELATED WORK

**Foundational Importance Metrics.** Early pruning methods use static metrics to assess parameter importance. Gradient-based approaches employ Taylor expansions, from first-order methods (Molchanov et al., 2017; Ma et al., 2023) to second-order Hessian-based improvements (Molchanov et al., 2019). These gradient-based methods typically adopt one-hot cross entropy criterion, modeling label prediction while ignoring potential predictions. In contrast, our information entropy criterion models holistic predictions by considering all potential predictions. LoRAPrune (Zhang et al., 2023) reduces costs by using LoRA gradients as importance proxies. Gradient-free alternatives include Wanda (Sun et al., 2023), which combines weight magnitudes with activation norms, and FLAP (An et al., 2024), which uses activation fluctuations. Reconstruction-based methods like SlimGPT (Ling et al., 2024) minimize post-pruning squared error, while SparseGPT (Frantar & Alistarh, 2023) frames pruning as layer-wise sparse regression for efficient one-shot pruning.

**Pruning Strategies.** Some methods employ dynamic sparsity allocation strategies. OWL (Yin et al., 2023) sets layer-specific sparsity based on outlier weight distributions, while SlimLLM (Guo et al., 2025) uses input-output cosine similarity for pruning ratios. Layer removal approaches like ShortGPT (Men et al., 2024) use "Block Influence" metrics to discard redundant layers, while Chen et al. (2025) replace such layers with lightweight MLPs. Other methods include APT (Zhao et al., 2024), which allocates rates based on layer sensitivity, and learnable approaches like Compresso (Guo et al., 2023) with mask generators and LoRAP (Li et al., 2024) with differentiated FFN/MHA strategies. After pruning, these approaches only minimize the change of label-related prediction, whereas our method minimizes the change of global prediction distribution.

**Combination with Other Techniques and Performance Recovery.** Structured pruning requires performance recovery to mitigate accuracy degradation. Existing methods present distinct trade-offs: Fine-tuning directly retrains pruned models, achieving reliable accuracy restoration but with high computational cost. Low-rank integration (Hsu et al., 2022; Yu & Wu, 2023) and knowledge distillation (Muralidharan et al., 2024) offer parameter efficiency and guided learning respectively, yet typically underperform compared to fine-tuning. Retraining-free approaches like Olica (He & Lin, 2025) use linear calibration for computational efficiency but provide limited recovery.

**Entropy-Based Pruning Criterion.** Some methods use information entropy as pruning criterion. NEPENTHE (Liao et al., 2024) removes weights from low-entropy activation layers, while DenoiseRotator (Gu et al., 2025) reshapes importance distributions through learned transformations, concentrating scores on fewer weights to minimize entropy and improve pruning effectiveness. Our information entropy criterion distinguishes itself by modeling holistic predictions and considering all potential predictions, enabling us to minimize the change of global prediction distribution after pruning with the advantage of a label-free criterion.

## 3 PRELIMINARY

In modern Transformer-based LLMs, MLP module within each block has evolved, often taking the form of a SwiGLU network.

$$\text{MLP}(x) = \left(\text{SiLU}(xW_{\text{gate}}) \odot (xW_{\text{up}})\right) W_{\text{down}}, \tag{1}$$

where the layer is configured without the bias term. The input is $x \in \mathbb{R}^{d_{\text{model}}}$. The weight matrices $W_{\text{gate}}, W_{\text{up}} \in \mathbb{R}^{d_{\text{model}} \times d_{\text{hidden}}}$, and $W_{\text{down}} \in \mathbb{R}^{d_{\text{hidden}} \times d_{\text{model}}}$. The hidden activation $h = \text{SiLU}(xW_{\text{gate}}) \odot (xW_{\text{up}})$ has dimensions $h \in \mathbb{R}^{d_{\text{hidden}}}$, and its individual components $h_i$ represent the hidden neurons, which we aim to prune.

To determine which neurons to prune, we estimate their importance by the effect their removal would have on the loss of model predictions, $\mathcal{L}$. We efficiently evaluate the effect of neuron removal using

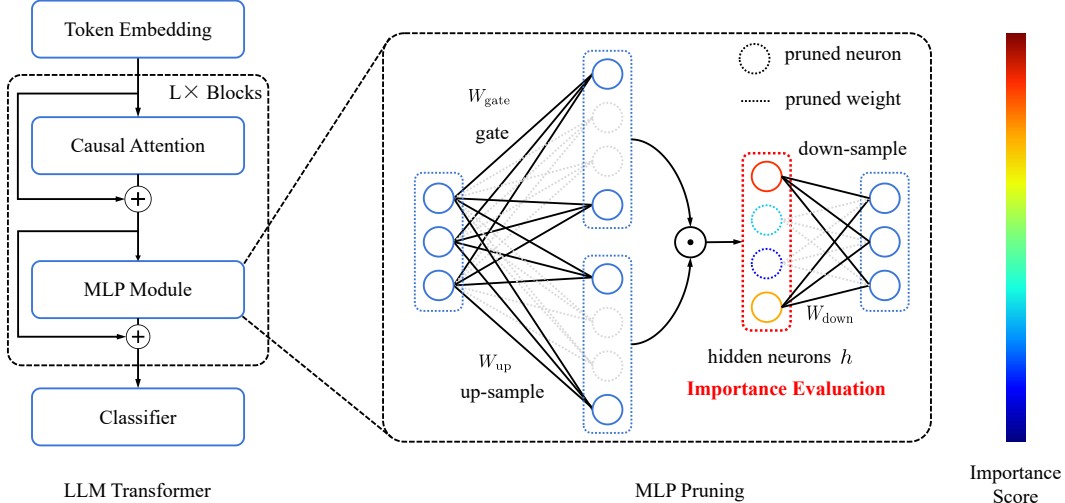

Figure 2: The overview of our high fidelity prune method. We focus on the pruning of hidden neurons $h$ of parameter-intensive MLP modules of LLM Transformer. To this end, we apply information entropy of model prediction to evaluate the importance scores of hidden neurons $h$ based on Taylor expansion. Then, we rank the hidden neurons $h$ by the obtained importance scores and prune the least important neurons, thus reducing model size while minimizing performance degradation.

first-order Taylor expansion. The change in loss, $\Delta\mathcal{L}_i$, from ablating the $i$-th neuron (i.e., setting its activation $h_i$ to zero) can be approximated as

$$\Delta\mathcal{L}_i = \mathcal{L}(h_i = 0) - \mathcal{L}(h_i) \approx \frac{\partial\mathcal{L}}{\partial h_i}(0 - h_i) = -\frac{\partial\mathcal{L}}{\partial h_i}h_i. \tag{2}$$

The magnitude of loss change, $\mathcal{I}(h_i) = |\Delta\mathcal{L}_i| = \left|\frac{\partial\mathcal{L}}{\partial h_i}h_i\right|$, is then used as the importance score of neuron $h_i$.

## 4 METHOD

### 4.1 OVERVIEW

In this paper, we propose HFPrune, a novel method that compress large language models by structurally pruning the Multi-Layer Perceptron (MLP) modules within their transformer architecture. The HFPrune process consists of three main stages. First, we establish a novel, label-free criterion to quantify neuron importance, the information entropy of the model's global prediction distribution as shown in Figure 1 (b). Unlike previous approaches that rely on one-hot cross entropy criterion and focus exclusively on modeling label prediction while ignoring all potential predictions, our method models holistic predictions by considering all potential predictions across the entire vocabulary. Then, we compute an importance score for each neuron by applying first-order Taylor expansion to this entropy-based criterion. Second, based on these scores, we prune the model by removing a fixed ratio of the least influential neurons from each MLP layer, as depicted in Figure 2. After pruning, while previous methods only minimize the change of label-related predictions, our approach minimizes the change of the global prediction distribution. Finally, a brief fine-tuning phase is employed to counteract any performance degradation and restore the model's capabilities.

### 4.2 INFORMATION ENTROPY FOR IMPORTANCE EVALUATION

Diverging from previous Taylor-based methods that rely on one-hot cross entropy criterion and focus solely on label prediction, our approach introduces an information entropy criterion for assessing neuron importance. Specifically, for a given input $x$, we define our criterion $\mathcal{C}_H(x)$, as the information entropy of the model's prediction distribution $P = \{p_1(x), p_2(x), \ldots, p_V(x)\}$ over a

---

**Algorithm 1** High Fidelity Pruning

---

1: **Input:** Pre-trained model $M$, calibration dataset $\mathcal{D}_{\text{calib}}$, MLP pruning ratio $\rho_{\text{mlp}}$.
2: **Output:** Pruned model $M'$.
3: **for** each MLP module $l$ in $M$ **do**
4:     Initialize importance scores $\{\mathcal{I}_i\}_i = 0$ for each neuron $i$ in module $l$.
5:     **for** each input $x \in \mathcal{D}_{\text{calib}}$ **do**
6:         Compute hidden activations $h(x)$ of MLP module $l$.
7:         Compute the information entropy of the model prediction $\mathcal{C}_H(x)$.
8:         Compute gradients $\nabla_{h(x)}\mathcal{C}_H(x) = \frac{\partial \mathcal{C}_H(x)}{\partial h(x)}$ by backward computation.
9:         **for** each neuron $i$ in hidden layer of MLP module $l$ **do**
10:            $\mathcal{I}_i \leftarrow \mathcal{I}_i + \left| \frac{\partial \mathcal{C}_H(x)}{\partial h_i(x)} h_i(x) \right|.$
11:         **end for**
12:     **end for**
13:     **for** each neuron $i$ in MLP module $l$ **do**
14:         $\mathcal{I}_i \leftarrow \frac{\mathcal{I}_i}{|\mathcal{D}_{\text{calib}}|}.$
15:     **end for**
16:     Let $k = \lfloor \rho_{\text{mlp}} \cdot d_{\text{hidden}} \rfloor.$
17:     Identify the set $K$ of indices of the $k$ neurons with the lowest scores in $\mathcal{I}$.
18:     Update MLP module $l$ by removing neurons with indices in $K$.
19:     Obtain the final model $M'$.
20: **end for**

---

vocabulary size of $V$ as

$$\mathcal{C}_H(x) = -\sum_{j=1}^{V} p_j(x) \log_2 p_j(x). \tag{3}$$

This information entropy criterion provides a holistic signal capturing the model's predictive confidence across the entire vocabulary, considering all potential predictions rather than focusing on specific labels. Moreover, our proposed information entropy criterion doesn't rely on the given labels for training, thus allowing label-free Taylor pruning. We now apply it on Taylor-based importance evaluation to quantify the importance of each hidden neuron. For input $x$, the importance of the $i$-th neuron, $\mathcal{I}_i(x)$, is the magnitude of the entropy change caused by ablating its activation, $h_i(x)$. To obtain a robust and generalized score, we average this value across a calibration dataset, $\mathcal{D}_{\text{calib}}$, to compute the final importance score, $\mathcal{I}_i(x)$ as

$$\mathcal{I}_i(x) = \frac{1}{|\mathcal{D}_{\text{calib}}|} \sum_{x \in \mathcal{D}_{\text{calib}}} \left| \frac{\partial \mathcal{C}_H(x)}{\partial h_i(x)} h_i(x) \right|. \tag{4}$$

A higher score $\mathcal{I}_i$ signifies that the neuron is more critical for preserving the fidelity of the model's global prediction distribution.

### 4.3 HIGH FIDELITY PRUNING

With the importance score $\mathcal{I}_i$ computed for every hidden neuron, we proceed to the structural pruning phase. This process is applied uniformly across all MLP layers. For each layer, we rank its hidden neurons by their importance scores and remove a fraction of those deemed least influential the ones with the lowest scores. The fraction of neurons to be pruned is controlled by a predefined ratio, $\rho_{mlp}$. Specifically, for a layer with $d_{\text{hidden}}$ neurons, we remove the $k = \lfloor \rho_{\text{mlp}} \cdot d_{\text{hidden}} \rfloor$ neurons with the lowest importance scores. This removal is achieved by modifying the weight matrices. Specifically, for each pruned neuron, we excise the corresponding row from the up-projection $W_{\text{up}}$ and gate-projection $W_{\text{gate}}$ matrices and the corresponding column from the down-projection $W_{\text{down}}$.

By pruning based on our information entropy criterion, we ensure that the resulting compressed model minimizes the change in global prediction distribution rather than just preserving label-specific predictions. This comprehensive approach maintains the model's distributional integrity across all potential outputs, leading to better preservation of the model's overall predictive capabilities. The overall algorithm can be summarized as Algorithm 1.

Table 1: Zero-shot performance on LLaMA-2-7B, which are finetuned on the LaMini dataset. The average accuracy is calculated among 10 zero-shot benchmarks. The **"bolded"** marked results denotes the best one under the same pruning ratio.

| Ratio | Method | ARCc | ARCe | BoolQ | Crows | OBQA | PIQA | Race | SiQA | TfQA | Wino | Average |
|-------|--------|------|------|-------|-------|------|------|------|------|------|------|---------|
| 0% | Llama-2-7B | 45.1 | 73.8 | 79.4 | 67.4 | 44.2 | 78.7 | 40.1 | 46.5 | 38.8 | 69.3 | 58.3 |
| 20% | LLM-pruner | 40.4 | 70.1 | 80.2 | 61.7 | 38.8 | 75.8 | 39.0 | 47.1 | 43.9 | 64.3 | 56.1 |
| | LoRAPrune | 41.6 | 71.0 | 81.7 | 58.7 | 41.4 | 76.7 | 40.4 | 44.0 | **65.9** | 65.9 | 56.7 |
| | LoRAP | 38.5 | 66.0 | 70.9 | – | 39.6 | **78.1** | – | – | – | 65.7 | – |
| | SDMPrune | 43.9 | 72.3 | 81.7 | **62.1** | 42.0 | 77.0 | 41.3 | 48.5 | 44.9 | **68.4** | 58.2 |
| | HFPrune (Ours) | **47.1** | **73.8** | **85.2** | 60.2 | **43.2** | 77.3 | **43.3** | **49.5** | 44.7 | 66.2 | **59.0** |
| 30% | LLM-pruner | 38.0 | 64.8 | 75.6 | **62.3** | 36.4 | 73.4 | 35.7 | 47.3 | 42.3 | 62.9 | 53.9 |
| | LoRAPrune | 38.6 | 65.1 | 74.1 | 61.4 | 37.4 | 72.9 | 39.0 | 46.3 | **44.8** | **66.5** | 54.6 |
| | LoRAP | 35.5 | 60.6 | 69.6 | – | 37.8 | **76.7** | – | – | – | 63.0 | – |
| | SDMPrune | 39.6 | 67.9 | 80.4 | 58.5 | 37.2 | 75.2 | **40.0** | 47.8 | 43.7 | 65.4 | 55.6 |
| | HFPrune (Ours) | **41.9** | **70.2** | **82.9** | 58.1 | **40.0** | 75.2 | 39.5 | **48.8** | 44.2 | 62.4 | **56.3** |

# 5 EXPERIMENT

## 5.1 EXPERIMENTAL SETUP

**Evaluation and Datasets.** To evaluate the effectiveness our method, we evaluate our proposed HFPrune on LLaMA and Qwen series models. Following prior work (Ma et al., 2023; Zhang et al., 2024), we assess the zero-shot performance of the pruned models using the lm_eval (Gao et al., 2021). Our evaluation encompasses a diverse set of zero-shot benchmarks, including ARC-easy (Clark et al., 2018), ARC-challenge (Clark et al., 2018), BOOLQ (Clark et al., 2019), Crows-Pairs (Nangia et al., 2020), OpenBookQA (Mihaylov et al., 2018),PIQA (Bisk et al., 2020), Race (Lai et al., 2017), SocialQA (Sap et al., 2019), TruthfulQA (Lin et al., 2021), and Winogrande (Sakaguchi et al., 2020).

**Calibration and Fine-Tuning Datasets.** We utilize two distinct datasets for the calibration and fine-tuning stage, respectively. In the calibration stage, we obtain importance score using a calibration dataset of 43,128 sequences randomly sampled from the C4 dataset (Raffel et al., 2020). Following the protocol of Wanda (Sun et al., 2023), each sequence is processed to a fixed length of 1,024 tokens, where longer sequences are randomly cropped into 1,024 tokens. In the fine-tuning stage, we use the LaMini-instruction dataset (Wu et al., 2024) across all experiments for fair comparison. This dataset consists of instruction-response pairs generated by GPT-3.5-turbo from various prompt resources.

**Implementation Details.** To recover performance after pruning, each model variant undergoes a brief fine-tuning stage. We fine-tune each model for 2 epochs on the LaMini-instruction dataset using LoRA strategy. This process employs the AdamW optimizer with BF16 precision and a cosine learning rate scheduler. Further implementation details, including specific hyperparameters, are available in the Section A.1 of appendix.

## 5.2 EXPERIMENTAL RESULTS

### 5.2.1 COMPARISON OF EXISTING METHODS

To facilitate a fair comparison against existing methods, we apply HFPrune to models from the LLaMA and Qwen series. This process yields a suite of model variants at different sparsity levels, enabling a thorough evaluation of the trade-off between compression and performance.

To validate the effectiveness of our method, we compare our method with existing structural pruning methods on LLaMA-2-7B model and then evaluate the pruned model on 10 popular zero-shot benchmarks. The results are listed in Table 1. The results demonstrate that our method significantly outperforms other structural pruning methods, when 20% and 30% parameters of LLaMA-2-7B model are pruned. When pruning ratio is set to 20%, our method achieves 59% average accuracy over 10 popular zero-shot benchmarks, surpassing the second best method, SDMPrune, by 0.8%. Notably, our method even outperforms the original model by 0.7%. We further explore high prun-

Table 2: Zero-shot performance on smaller LLMs, including LLaMA3.2-3.2B (left) and LLaMA3.2-1.2B (right) models. The settings are identical to the ones of LLaMA-2-7B. More detailed results on each benchmark are available in the Appendix.

(a) LLaMA3.2-3.2B Results

| Prune% | Method | #Params | Avg Zero-shot A. ↑ |
|---|---|---|---|
| - | - | 3.2B | 54.30 |
| 20% | LLMPruner | 2.56B | 52.24 |
| | LoRAPrune | 2.57B | 50.30 |
| | SDMPrune | 2.56B | 53.37 |
| | HFPrune(Ours) | 2.56B | **54.07** |
| 30% | LLMPruner | 2.25B | 48.68 |
| | LoRAPrune | 2.25B | 48.36 |
| | SDMPrune | 2.25B | 51.03 |
| | HFPrune(Ours) | 2.25B | **52.28** |

(b)LLaMA3.2-1.2B Results

| Prune% | Method | #Params | Avg Zero-shot A. ↑ |
|---|---|---|---|
| - | - | 1.2B | 51.47 |
| 20% | LLMPruner | 0.989B | 47.39 |
| | LoRAPrune | 0.989B | 47.17 |
| | SDMPrune | 0.989B | 48.94 |
| | HFPrune(Ours) | 0.989B | **50.77** |
| 30% | LLMPruner | 0.865B | 44.79 |
| | LoRAPrune | 0.865B | 44.86 |
| | SDMPrune | 0.865B | 46.40 |
| | HFPrune(Ours) | 0.865B | **48.59** |

ing ratio as 30%, our HFPrune is consistently superior to other methods. The above results indeed support the superiority of our method on LLM compression task.

To further verify the generalization on smaller scale LLMs, our proposed HFPrune is also applied on LLaMA3.2-3.2B and LLaMA3.2-1.2B models. The results shown in Table 2 also demonstrate that our method can effectively reduce the performance gap between the pruned model and the original model.

Table 3: Zero-shot performance on Qwen series models. The settings are identical to the ones of LLaMA-2-7B.

| Model | Ratio | Method | ARCc | ARCe | BoolQ | Crows | OBQA | PIQA | Race | SiQA | TfQA | Wino | Average |
|---|---|---|---|---|---|---|---|---|---|---|---|---|---|
| Qwen2.5-7B | 0% | Original | 51.1 | 77.5 | 84.7 | 64.0 | 46.8 | 79.8 | 41.4 | 54.8 | 56.4 | 73.2 | 63.0 |
| | 20% | SDMPrune | 48.1 | 69.7 | **84.6** | **64.2** | 44.4 | 77.6 | **41.7** | 48.0 | 51.8 | **67.2** | 59.7 |
| | | HFPrune(Ours) | **52.1** | **74.5** | 84.6 | 62.9 | **45.4** | **78.1** | 41.2 | **48.1** | **52.0** | 64.8 | **60.4** |
| | 30% | SDMPrune | 40.5 | 63.6 | 81.1 | **62.2** | 38.4 | 74.2 | 39.4 | 44.5 | **49.1** | 60.2 | 55.3 |
| | | HFPrune(Ours) | **48.2** | **70.3** | **83.2** | 60.4 | **41.6** | **76.1** | **40.7** | **48.9** | 48.2 | **62.8** | **58.0** |
| Qwen2.5-1.5B | 0% | Original | 45.4 | 72.0 | 73.0 | 60.9 | 40.4 | 76.0 | 36.5 | 48.9 | 46.6 | 63.5 | 56.3 |
| | 20% | SDMPrune | 32.3 | 59.2 | 72.1 | **56.2** | 35.2 | 72.0 | 37.7 | 43.6 | **44.7** | 58.2 | 51.1 |
| | | HFPrune(Ours) | **41.8** | **68.8** | **79.4** | 55.3 | **39.4** | **74.1** | **38.7** | **46.4** | 42.2 | **59.8** | **54.6** |
| | 30% | SDMPrune | 26.3 | 49.5 | 62.9 | **55.0** | 32.8 | 68.6 | 34.9 | 42.1 | 43.8 | 57.9 | 47.4 |
| | | HFPrune(Ours) | **35.2** | **62.6** | **76.0** | 54.0 | **34.6** | **70.8** | **36.2** | **44.0** | **44.5** | **58.6** | **51.7** |
| Qwen3-1.7B | 0% | Original | 42.8 | 70.0 | 77.6 | 55.2 | 37.8 | 71.9 | 36.8 | 44.9 | 46.0 | 61.6 | 54.4 |
| | 20% | SDMPrune | 31.3 | 58.5 | 70.8 | 53.7 | 33.4 | 71.4 | 37.1 | 43.8 | 44.7 | **58.6** | 50.3 |
| | | HFPrune(Ours) | **39.1** | **69.4** | **78.9** | **55.8** | **36.2** | **72.4** | **39.7** | **46.4** | **46.4** | 58.2 | **54.3** |
| | 30% | SDMPrune | 25.9 | 45.1 | 62.3 | 55.1 | 29.8 | 66.2 | 35.3 | 40.5 | 43.4 | **56.7** | 46.0 |
| | | HFPrune(Ours) | **34.5** | **61.9** | **78.2** | **55.5** | **33.8** | **70.8** | **36.3** | **44.1** | **45.2** | 56.4 | **51.7** |

We also assess the effectiveness of our method on Qwen series models, including Qwen2.5-1.5B, Qwen2.5-7B and Qwen3-1.7B. For brief, we focus on the comparative experiments with the previous best methods, SDMPrune. The results are reported in Table 3. Our method consistently surpasses SDMPrune across various model sizes and pruning ratios. We analyze the potential reason as follows. Despite SDMPrune aims to exploit full prediction distribution, it still relies on Taylor pruning based on one-hot cross entropy criterion in the initial stage due to zero-gradient issue of self distillation. SDMPrune fails to capture holistic prediction distribution during importance evaluation, thus leading to inferior performance. On the contrary, our method considers all potential predictions during Taylor-based importance evaluation, thus providing more accurate pruning criterion.

### 5.2.2 ACCELERATION PERFORMANCE OF PRUNED MODELS

We evaluate the practical acceleration benefits of HFPrune on the pruned LLaMA2-7B models, with results presented in Table 4. Prefill latency was measured using an input sequence of 256 tokens and a single-token generation task. To measure decoding throughput, we set the batch size to 8 and recorded the latency for generating 256 tokens. The results demonstrate a direct correlation between pruning and efficiency. As the pruning ratio increases, both model parameters and prefill latency decrease significantly. Notably, pruning 30% of the MLP layers results in a 1.47× speedup in prefill latency, confirming a tangible inference advantage.

Table 4: Acceleration results of the structural pruned LLMs. Prefill Latency and decoding throughput were measured on a single NVIDIA A6000 GPU without employing any acceleration frameworks.

| Model | #Params | Ratio | Avg zero-shot ↑ | Prefill (ms) ↓ | Speedup | Decoding (tokens/s) ↑ |
|---|---|---|---|---|---|---|
| | 6.7B | 0 | 58.3 | 57.5 | 1.00× | 473.9 |
| LLaMA2-7B | 5.4B | 20% | 59.0 | 46.3 | 1.24× | 553.9 (+17.9%) |
| | 4.7B | 30% | 56.3 | 42.1 | 1.35× | 593.8 (+25.3%) |

Beyond the inference speed of pruned model, we also assess the computational efficiency of the pruning process itself, comparing HFPrune against SDMPrune (Zhu & Shen, 2025). As detailed in Table 5, HFPrune is substantially more efficient in both time and memory usage. For instance, when pruning LLaMA2-7B, HFPrune is approximately 3× faster than SDMPrune and consumes 31% less peak GPU memory.

Table 5: Efficiency comparison of the pruning process. The metrics are measured using 1000 randomly sampled sequences from the C4 dataset, each with a length of 1024 tokens. All models were run on four A6000 GPUs.

| Model | Method | Elapsed Time (s) | Peak Memory (GB) |
|---|---|---|---|
| LLaMA3.2-1.2B | HFPrune (Ours) | 150.7 | 7.5 |
| | SDMPrune | 317.1 | 12.9 |
| LLaMA3.2-3.2B | HFPrune (Ours) | 305.8 | 17.3 |
| | SDMPrune | 825.2 | 25.7 |
| LLaMA2-7B | HFPrune (Ours) | 508.9 | 35.3 |
| | SDMPrune | 1539.8 | 51.2 |

## 5.3 ABLATION STUDY

### 5.3.1 THE EFFECT OF INFORMATION ENTROPY CRITERION

To isolate and validate the standalone effectiveness of our proposed neuron importance criterion, we conduct a direct comparison on the LLaMA-2-7B model. Our information entropy (IE) criterion is benchmarked against two key alternatives: the cross entropy (CE) loss criterion and the self-distillation (SD) loss criterion. Crucially, all comparisons are performed without any post-pruning fine-tuning. The results in Table 6 demonstrate clear superiority of our IE criterion. As pruning ratio increases from 20% to 30%, our method maintains relatively stable performance degradation compared to baseline methods, indicating superior robustness of our importance estimates. Our method achieves the highest average performance at both pruning ratios, outperforming the best baselines 0.5 percentage points respectively. This outperformance in a retraining-free scenario confirms our central hypothesis: by modeling holistic predictions rather than focusing solely on label-related predictions, our IE criterion provides fundamentally more accurate measures of neuron importance, yielding superior models immediately after pruning.

Table 6: Zero-Shot Performance Comparison of Different Pruning Criteria on LLaMA-2-7B (No Fine-Tuning). Where IE (Information Entropy) is our proposed criterion, while SD (Self-Distillation) and CE (Cross-Entropy) represent the baseline loss criteria.

| Ratio | Criterion | ARCc | ARCe | BoolQ | Crows | OBQA | PIQA | Race | SiQA | TfQA | Wino | Avg |
|---|---|---|---|---|---|---|---|---|---|---|---|---|
| | CE Loss | **37.2** | 62.3 | 71.3 | 58.9 | 39.4 | 74.5 | 36.9 | 43.1 | 36.3 | **65.9** | 52.6 |
| 20% | SD Loss | 34.2 | **63.6** | 67.1 | 59.6 | 38.4 | **75.0** | 37.4 | **44.0** | 36.6 | 62.8 | 51.9 |
| | IE (ours) | 37.0 | 63.4 | **71.8** | **60.0** | **40.0** | 74.9 | 37.7 | 43.7 | **37.0** | 65.0 | **53.1** |
| | CE Loss | **31.4** | 49.0 | 63.1 | 54.1 | 32.4 | 69.4 | 32.8 | 40.9 | 37.2 | **57.4** | 46.8 |
| 30% | SD Loss | 28.2 | **50.3** | 49.0 | 53.9 | 32.6 | 69.3 | **33.6** | 41.5 | 37.3 | 56.4 | 45.2 |
| | IE (ours) | 30.1 | 49.4 | **65.8** | **54.6** | 33.2 | **70.2** | 33.2 | **41.7** | **37.5** | 57.1 | **47.3** |

### 5.3.2 EFFECT OF PRUNING CRITERION ON OUTPUT DISTRIBUTION

To quantitatively validate that our information entropy (IE) criterion better preserves the global prediction distribution than the cross-entropy (CE) criterion, we analyzed the pruned Llama2-7B model over 5,000 prompts from the C4 dataset. The results, presented in Table 7, confirm our method's superior fidelity across two key metrics. First, our IE criterion achieves a consistently lower Jensen-Shannon (JS) Distance. While the improvement at 20% sparsity is modest, the advantage becomes

more pronounced at the aggressive 30% ratio. This trend highlights that our method's ability to preserve the distribution's overall shape is most critical in high-compression scenarios. Second, the Top-15 Jaccard Similarity results provide complementary evidence, showing our approach better maintains the original model's most probable next tokens at both sparsity levels. Qualitative examples that visually illustrate these findings are provided in Appendix A.3.

Table 7: Comparison of pruning methods based on output distribution similarity.

| Ratio | Criterion | JS Distance (↓) | Top-15 Jaccard (↑) |
|---|---|---|---|
| 20% | CE Loss | 0.243 | 0.439 |
| | Ours (IE) | 0.241 | 0.445 |
| 30% | CE Loss | 0.362 | 0.588 |
| | Ours (IE) | 0.353 | 0.595 |

### 5.3.3 PRUNE DIFFERENT PARTS

To validate our hypothesis of exclusively targeting MLP modules, we compare pruning only MLP modules versus pruning both attention and MLP modules. The results in Table 8 provide decisive evidence favoring the MLP-only strategy. MLP-only pruning consistently outperforms attention&MLP pruning across both scenarios. After fine-tuning, MLP-only achieves 61.9% average performance at 20% pruning compared to 60.3% for attention&MLP pruning. This gap widens at 30% pruning, demonstrating superior scalability. The fine-tuning gains reveal different recovery patterns. MLP-only pruning shows larger improvement potential, while attention&MLP pruning exhibits more limited recovery. This suggests MLP modules contain more recoverable, distributed knowledge. Therefore, this experiment provides evidence that focusing pruning efforts on MLP modules is a more effective and robust strategy for compressing LLMs.

Table 8: Prune different parts on LLaMA-2-7B

| Ratio | Method | ARCc | ARCe | BoolQ | Crows | OBQA | PIQA | Race | SiQA | Wino | Average |
|---|---|---|---|---|---|---|---|---|---|---|---|
| 20% | attn&mlp w/o tune | 40.0 | 69.0 | 65.1 | 60.0 | 40.0 | 76.0 | 32.5 | 41.0 | 55.0 | 53.2 |
| | attn&mlp w/ tune | 48.3 | 73.9 | 84.6 | **60.5** | 42.8 | 76.7 | 41.9 | 49.1 | 65.0 | 60.3 |
| | mlp w/o tune | 37.0 | 63.4 | 71.8 | 60.0 | 40.0 | 74.9 | 37.7 | 43.7 | 65.0 | 54.8 |
| | mlp w/ tune | **50.3** | **75.2** | **85.8** | 60.0 | **45.2** | **77.3** | **44.5** | **50.5** | **67.9** | **61.9** |
| 30% | attn&mlp w/o tune | 33.0 | 57.0 | 56.0 | 54.0 | 33.4 | 70.0 | 24.2 | 38.1 | 52.3 | 46.4 |
| | attn&mlp w/ tune | 44.6 | 73.0 | 82.9 | 56.3 | 41.4 | 76.0 | 39.9 | 48.1 | 60.1 | 58.0 |
| | mlp w/o tune | 30.1 | 49.4 | 65.8 | 54.6 | 33.2 | 70.2 | 33.6 | 41.7 | 57.1 | 48.4 |
| | mlp w/ tune | **47.6** | **73.4** | **83.9** | **60.3** | **43.2** | **76.7** | **42.2** | **49.6** | **63.2** | **60.0** |

## 6 CONCLUSION

In this paper, we present HFPrune, a novel structured pruning method for large language models that overcomes a fundamental limitation of existing Taylor-based approaches. Our core innovation is replacing the one-hot cross entropy criterion with the information entropy of the model's global prediction distribution as the criterion for assessing neuron importance. This entropy-based criterion offers distinct advantages. By modeling holistic predictions, it provides a more faithful measure of a neuron's contribution to the model's overall predictive fidelity. The pruning process is thus guided to minimize the change of the global prediction distribution, preserving intrinsic knowledge far more effectively than methods that only focus on a single target token. Furthermore, our approach avoids the computational overhead and gradient initialization issues inherent in self distillation methods while achieving superior pruning results. Looking forward, our work opens several promising avenues for research. The entropy-based importance metric could be extended to other model compression techniques, such as quantization. Additionally, future investigations could explore its application to diverse neural network architectures or the development of adaptive pruning ratios based on layer specific entropy, potentially yielding even greater gains in compression efficiency.

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

## A APPENDIX

### A.1 IMPLEMENTATION DETAILS.

To restore model performance post-pruning, we conduct a brief fine-tuning phase using the AdamW optimizer and BF16 precision. As detailed in Table 9, we customized the hyperparameters for each model and sparsity ratio. For instance, larger models like LLaMA2-7B and Qwen2.5-7B were trained with a larger batch size of 512 on 8×A6000 GPUs, while smaller models used a batch size of 256 on 4×4090 GPUs. The learning rates were also adjusted, ranging from $2 \times 10^{-4}$ to $4 \times 10^{-4}$ depending on the model scale.

Table 9: Implementation Details of the structural pruned LLMs.

| Model | Ratio | $\rho_{mlp}$ | Pack | lr | beta1 | beta2 | Weight decay | Precision | Batch size | Device |
|---|---|---|---|---|---|---|---|---|---|---|
| LLaMA3.2-1.2B | 20% | 0.70 | yes | $4 \times 10^{-4}$ | 0.9 | 0.999 | 0 | BF16 | 256 | 4×4090 |
| | 30% | 0.55 | yes | $4 \times 10^{-4}$ | 0.9 | 0.999 | 0 | BF16 | 256 | 4×4090 |
| LLaMA3.2-3.2B | 20% | 0.70 | yes | $4 \times 10^{-4}$ | 0.9 | 0.999 | 0 | BF16 | 256 | 4×4090 |
| | 30% | 0.55 | yes | $4 \times 10^{-4}$ | 0.9 | 0.999 | 0 | BF16 | 256 | 4×4090 |
| LLaMA2-7B | 20% | 0.70 | yes | $2 \times 10^{-4}$ | 0.9 | 0.999 | 0 | BF16 | 512 | 8×A6000 |
| | 30% | 0.54 | yes | $2 \times 10^{-4}$ | 0.9 | 0.999 | 0 | BF16 | 512 | 8×A6000 |
| Qwen2.5-1.5B | 20% | 0.74 | yes | $4 \times 10^{-4}$ | 0.9 | 0.999 | 0 | BF16 | 256 | 4×4090 |
| | 30% | 0.60 | yes | $4 \times 10^{-4}$ | 0.9 | 0.999 | 0 | BF16 | 256 | 4×4090 |
| Qwen3-1.7B | 20% | 0.68 | yes | $4 \times 10^{-4}$ | 0.9 | 0.999 | 0 | BF16 | 256 | 4×4090 |
| | 30% | 0.52 | yes | $4 \times 10^{-4}$ | 0.9 | 0.999 | 0 | BF16 | 256 | 4×4090 |
| Qwen2.5-7B | 20% | 0.74 | yes | $2 \times 10^{-4}$ | 0.9 | 0.999 | 0 | BF16 | 512 | 8×A6000 |
| | 30% | 0.61 | yes | $2 \times 10^{-4}$ | 0.9 | 0.999 | 0 | BF16 | 512 | 8×A6000 |

## A.2 MORE DETAILED EVALUATION RESULTS

To complement the summarized results presented in the main body, this section provides a more granular breakdown of our method's performance on the LLaMA-3.2 model series. Table 10 and Table 11 present the detailed zero-shot evaluation scores for the LLaMA-3.2-1B and LLaMA-3.2-3B models, respectively. These tables show the performance across benchmarks, offering a comprehensive view that reinforces the consistent advantage of our entropy-based pruning criterion over existing baselines at both 20% and 30% sparsity ratios.

Table 10: Zero-shot performance of the compressed LLaMA-3.2-1B. The **"bolded"** represents the best result under the same pruning ratio.

| Ratio | Method | ARCc | ARCe | BoolQ | Crows | OBQA | PIQA | Race | SiQA | TfQA | Wino | Average |
|-------|--------|------|------|-------|-------|------|------|------|------|------|------|---------|
| 0% | Llama-3.2-1.2B | 37.1 | 60.7 | 64.0 | 62.6 | 37.6 | 74.2 | 37.6 | 42.9 | 37.7 | 60.4 | 51.5 |
| 20% | LLM-pruner | 29.6 | 51.0 | 62.1 | 57.5 | **33.6** | 67.2 | 33.7 | 40.9 | 41.7 | 56.6 | 47.4 |
| | Compresso | 27.1 | 48.5 | 59.1 | **58.6** | 27.4 | 67.0 | 33.2 | 40.2 | **43.7** | 55.6 | 46.1 |
| | LoRAPrune | 29.6 | 51.0 | 66.2 | 54.4 | 29.6 | 66.4 | 34.1 | 42.0 | 41.7 | **56.9** | 47.2 |
| | SDMPrune | 31.1 | 55.2 | 67.6 | 57.8 | 32.4 | 70.3 | 35.4 | 42.7 | 40.2 | 56.6 | 48.9 |
| | HFPrune(Ours) | **36.0** | **60.9** | **72.8** | 55.4 | 33.0 | **71.3** | 36.9 | **43.6** | 41.3 | 56.6 | **50.8** |
| 30% | LLM-pruner | 27.3 | 46.3 | 60.2 | 54.3 | 27.4 | 64.0 | 32.9 | 38.4 | **42.5** | 54.6 | 44.8 |
| | Compresso | 25.7 | 45.3 | 61.8 | 55.8 | 26.5 | 62.3 | 31.8 | 37.1 | 40.7 | 53.5 | 44.1 |
| | LoRAPrune | 28.1 | 46.4 | 59.6 | 56.3 | 27.6 | 64.9 | 32.4 | 37.3 | 41.6 | 54.4 | 44.9 |
| | SDMPrune | 28.5 | 47.5 | 64.7 | **56.9** | 29.0 | 66.3 | 33.2 | 40.8 | 42.4 | 54.7 | 46.4 |
| | HFPrune(Ours) | **32.3** | **54.6** | **70.3** | 55.9 | **29.8** | **68.1** | 35.6 | 42.6 | 40.4 | **56.3** | **48.6** |

Table 11: Zero-shot performance of the compressed llama-3.2-3B. The **"bolded"** represents the best result under the same pruning ratio.

| Ratio | Method | BoolQ | Crows | OBQA | PIQA | Race | SiQA | TfQA | Average |
|-------|--------|-------|-------|------|------|------|------|------|---------|
| 0% | llama-3.2-3B | 73.7 | 63.3 | 40.6 | 77.9 | 38.9 | 46.7 | 39.2 | 54.3 |
| 20% | LLMPruner | 74.7 | 58.9 | 33.8 | 72.1 | **40.1** | 43.7 | 42.6 | 52.2 |
| | LoRAPrune | 68.8 | 57.9 | 33.1 | 71.4 | 35.3 | 42.9 | 42.6 | 50.3 |
| | SDMPrune | 75.3 | **59.3** | 36.2 | 73.6 | 38.9 | 44.6 | **45.8** | 53.4 |
| | HFPrune(Ours) | **79.8** | 58.8 | **37.8** | **74.6** | 39.0 | **46.0** | 42.6 | **54.1** |
| 30% | LLMPruner | 66.2 | 54.9 | 32.8 | 66.2 | 37.1 | 41.3 | 42.3 | 48.7 |
| | LoRAPrune | 65.4 | 53.9 | **33.8** | 69.5 | 34.0 | 41.2 | 40.7 | 48.4 |
| | SDMPrune | 72.8 | **58.1** | 30.4 | 70.7 | 36.8 | 43.3 | **45.1** | 51.0 |
| | HFPrune(Ours) | **77.6** | 57.3 | **33.8** | **71.0** | **38.6** | **46.5** | 41.3 | **52.3** |

In addition to the main results which include a performance recovery stage, we provide a crucial study to assess the standalone effectiveness of our proposed pruning criterion. Table 12 presents the zero-shot performance of the Qwen series models immediately after pruning and without any fine-tuning. This retraining-free evaluation isolates the impact of the importance scoring itself. The results show that our entropy-based method consistently preserves higher accuracy than the baseline across various models and sparsity levels, validating our central hypothesis that our criterion is more effective at preserving a model's intrinsic knowledge.

## A.3 QUALITATIVE ANALYSIS: DISTRIBUTION VISUALIZATION EXAMPLES

To visually illustrate the quantitative findings on distribution fidelity presented in the main text (Section 5.3.2), this section provides qualitative examples for four sample prompts. As shown in Figures 3, 4, 5, and 6, visualize and compare the next-token prediction distributions of three models: the original dense model, the model pruned with our Information Entropy (IE) criterion, and the model pruned with the traditional Cross-Entropy (CE) criterion. These examples visually confirm our quantitative results. A direct comparison shows that the distribution from the IE-pruned model

Table 12: Zero-shot performance of the compressed Qwen without any fine-tuning. The **"bolded"** represents the best result under the same pruning ratio.

| Model | Ratio | Method | ARCc | ARCe | BoolQ | Crows | OBQA | PIQA | Race | SiQA | TfQA | Wino | Average |
|---|---|---|---|---|---|---|---|---|---|---|---|---|---|
| Qwen2.5-7B | 0% | Dense | 51.1 | 77.5 | 84.7 | 64.0 | 46.8 | 79.8 | 41.4 | 54.8 | 56.4 | 73.2 | 63.0 |
| | 20% | SDMPrune | 37.5 | 58.5 | **74.3** | 61.2 | 38.7 | 76.1 | 37.1 | **50.4** | **50.2** | 63.5 | **54.8** |
| | | HFPrune(Ours) | **39.9** | **66.9** | 68.2 | **61.8** | **39.8** | **77.5** | **37.2** | 48.2 | 46.1 | 60.1 | 54.6 |
| | 30% | SDMPrune | 30.0 | 52.8 | **66.4** | **59.3** | 32.8 | 71.2 | 33.6 | 42.6 | **44.9** | **58.9** | 49.2 |
| | | HFPrune(Ours) | **34.0** | **58.3** | 61.9 | 58.3 | **35.2** | **74.1** | **35.2** | 44.3 | 43.1 | 56.2 | **50.1** |
| Qwen2.5-1.5B | 0% | Dense | 45.4 | 72.0 | 73.0 | 60.9 | 40.4 | 76.0 | 36.5 | 48.9 | 46.6 | 63.5 | 56.3 |
| | 20% | SDMPrune | 30.6 | 55.6 | **65.5** | 54.5 | 32.2 | 70.2 | 32.9 | 41.0 | **43.5** | 56.7 | 48.3 |
| | | HFPrune(Ours) | **32.3** | **56.8** | 64.0 | **56.1** | **34.0** | **71.9** | **34.7** | 42.6 | 41.1 | 57.0 | **49.1** |
| | 30% | SDMPrune | 23.2 | 40.5 | 57.9 | **54.4** | 28.8 | 65.6 | 32.3 | 38.5 | **44.2** | 54.8 | 44.0 |
| | | HFPrune(Ours) | **25.9** | **46.8** | 62.0 | 50.8 | **31.0** | **69.0** | 32.3 | 39.9 | 42.1 | 53.7 | **45.3** |
| Qwen3-1.7B | 0% | Dense | 42.8 | 70.0 | 77.6 | 55.2 | 37.8 | 71.9 | 36.8 | 44.9 | 46.0 | 61.6 | 54.4 |
| | 20% | SDMPrune | 23.3 | 36.5 | 51.0 | 49.6 | 26.4 | 63.2 | 32.1 | 36.6 | **43.9** | 54.1 | 41.7 |
| | | HFPrune(Ours) | **27.8** | **44.2** | **64.6** | **51.8** | **28.6** | **65.2** | **34.4** | **38.6** | 42.0 | **54.5** | **45.2** |
| | 30% | SDMPrune | **21.0** | 32.1 | 39.1 | 47.9 | **25.6** | 58.5 | **28.7** | 35.2 | **46.0** | **51.1** | 38.5 |
| | | HFPrune(Ours) | 20.0 | **35.3** | **62.1** | 47.9 | 25.2 | **59.7** | 27.1 | **35.5** | 43.8 | 50.5 | **40.7** |

more closely mirrors the original, both in its overall shape (corresponding to lower JS Distance) and in its preservation of high probability tokens (corresponding to higher Jaccard Similarity). This demonstrates the superior ability of our IE criterion to maintain the model's predictive fidelity.

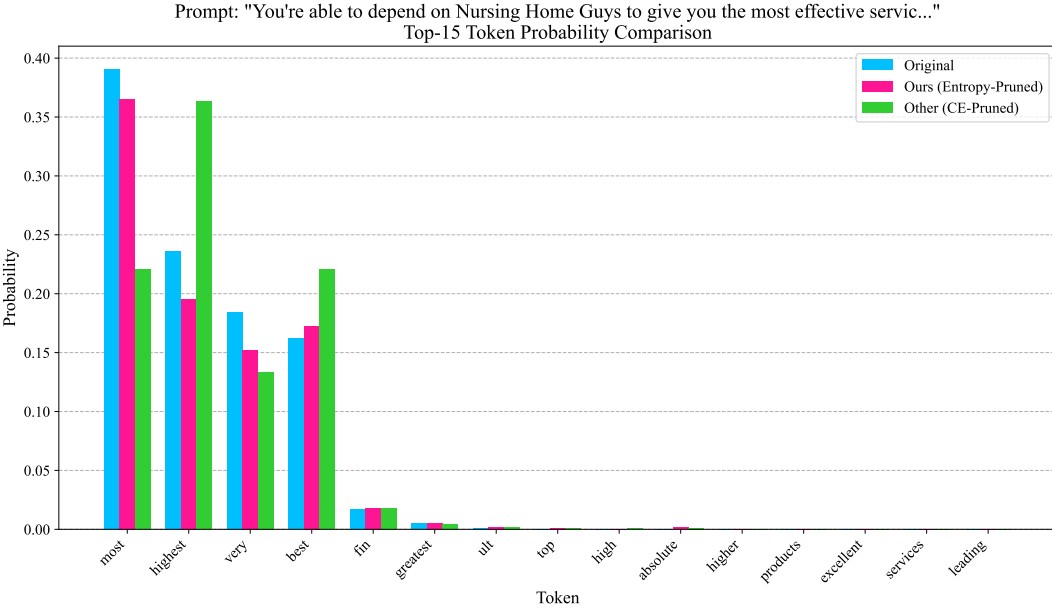

Figure 3: Sample1, You are able to depend on Local Bathroom Remodel Crew to deliver the very best expert services when it comes to Bathroom Remodeling in Williamstown, NJ. Our crew of experienced experts will provide the expert services that you require with the most innovative technologies around. We make sure

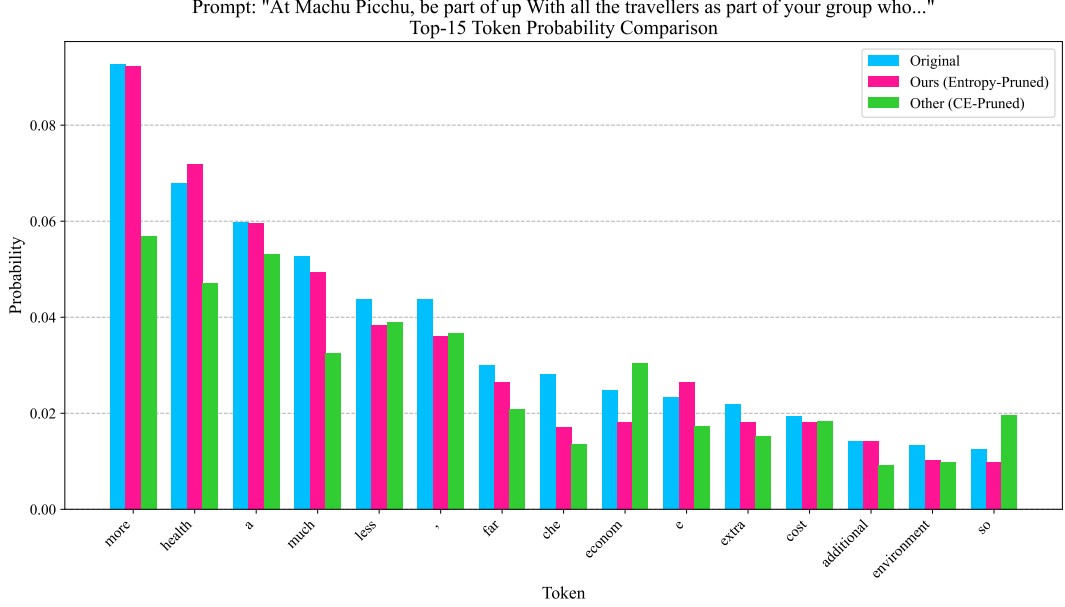

Figure 4: Sample2, At Machu Picchu, be part of up With all the travellers as part of your group who hiked the common Inca Trail. If skies are very clear, get pleasure from a spectacular views more than The traditional metropolis from your Sunlight Gate, right before going on a guided stroll round the ruins. Mobile Massage Can Save The Environment! A more pure, not to say

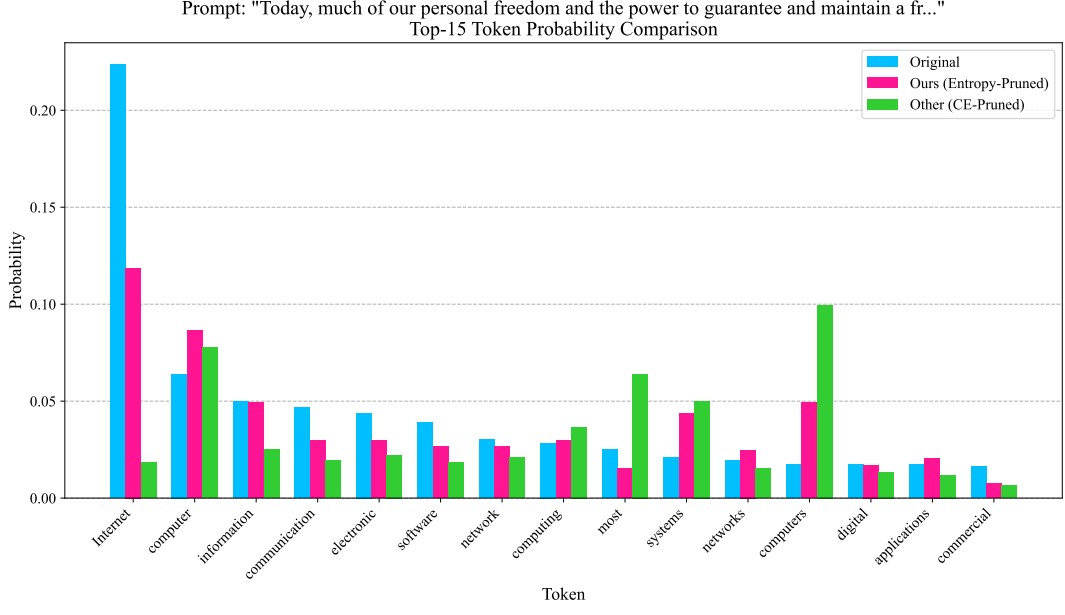

Figure 5: Sample3, Today, much of our personal freedom and the power to guarantee and maintain a free society depends on cryptographic primitives (e.g., digital signatures and encryption) incorporated in the security protocols of today's

## A.4 LLM USAGE

We utilized Large Language Model to refine the phrasing of this paper. The core contributions of this work including the motivation, core idea, method design, and experimental design are entirely the work of the authors.

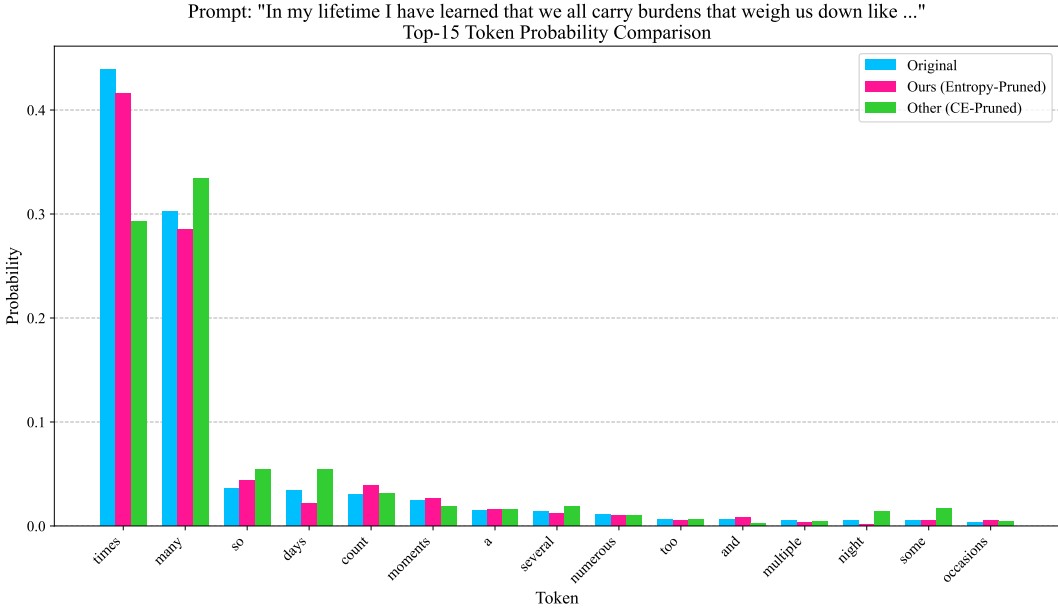

Figure 6: Sample4, In my lifetime I have learned that we all carry burdens that weigh us down like rocks threatening to squash us like bugs. I have also learned that the only true way to feel that burden lightened, to feel strong and capable of enduring is through the Lord Jesus Christ's Atonement. He suffered not only for our sins but for our physical pains, our doubts, our fears and regrets. Every burden we carry he has felt and taken upon his back. There have been

