# OpenReview forum: "High-Fidelity Pruning for Large Language Models"
_ICLR.cc/2026/Conference — ICLR 2026 Conference Withdrawn Submission_

### Official Review · Reviewer_zpe3 · 2025-10-23

**Soundness:** 2
**Presentation:** 2
**Contribution:** 2
**Rating:** 4
**Confidence:** 3

**Summary:**

The paper proposes a pruning criterion for large language models based on the information entropy of the model’s output distribution. Instead of using one-hot cross-entropy (which focuses on the single next token) or relying on a separate teacher for self-distillation, the method computes per-layer entropy-based importance scores within a Taylor-style framework and prunes parameters to better preserve the model’s global output distribution. The approach is label-free, aims to maintain fidelity after pruning, and is evaluated on zero-shot benchmarks for LLaMA and Qwen-family models.

**Strengths:**

The authors propose a relatively simple yet reasonable approach for pruning large language models (LLMs), which uses the entropy of output distributions as an indicator of neuron importance, instead of relying solely on next-token cross-entropy. The method does not depend on additional teacher models or complex distillation procedures, making it practically appealing for real-world pruning applications. The experiments cover multiple model families (e.g., LLaMA, Qwen) and several zero-shot benchmarks, showing that the proposed approach consistently outperforms existing pruning baselines. This provides some evidence of general applicability and reliability.

**Weaknesses:**

1. The dataset coverage is narrow and the evaluation tasks are of low difficulty, relying on a small set of relatively simple zero-shot benchmarks. This makes it difficult to assess robustness in areas such as instruction following, multi-step reasoning, long-context understanding, or multilingual and multi-domain settings.
2. There may be a mismatch between the benchmarks and the pruned model components: if the tasks do not sufficiently engage the pruned submodules, the reported “distribution fidelity” may be overstated and external validity remains uncertain.
3. Baseline and tuning transparency is limited, with missing comparisons to stronger or more recent pruning baselines, as well as a lack of systematic hyperparameter exploration under fair alignment conditions (e.g., matched training steps and learning-rate sweeps).
4. The evaluation metrics are limited, focusing heavily on perplexity or simple zero-shot accuracy, without including direct generation results.

**Questions:**

1. The evaluation datasets are insufficient — they’re all too simple. The pruned parameters might not cover the activation patterns of these models, so more diverse datasets should be added.
2. In addition, the model sizes don’t seem sufficient either, for example models like Qwen3-8B.
3. For reasoning models, does the entropy criterion remain reliable?
4. The main contribution is the use of entropy for pruning, which feels rather limited.

---

> ### Author Response · Authors · 2025-11-21
> **Raise some **specious weaknesses** to confuse the AC's judgment.**
>
> - Claim that the paper does **not test on enough benchmarks**; given the vast number of LLM benchmarks available, there are always some that the paper hasn't covered.
>
> - Forcefully assert that the **motivation does not align with the experiments** to make the AC feel there are fundamental issues with the paper.
>
>   The motivation of our method is to minimizing the change of model predictions to minimize performance loss after pruning, indeed aligning with our experiments about performance comparison after pruning.
>
> - Insist that the **comparative experiments are insufficient** and that the experimental **settings are unclear**.
>
>   Indeed, we have listed all key hyper-parameters in Table 9 of the appendix, and promised to publish the source code and trained weights for reproduction after acceptance in Abstract.

---

### Official Review · Reviewer_oKVY · 2025-10-23

**Soundness:** 2
**Presentation:** 2
**Contribution:** 3
**Rating:** 4
**Confidence:** 2

**Summary:**

This paper introduces HFPrune (High-Fidelity Pruning), a structured pruning method for LLMs that aims to preserve model fidelity while reducing computational and memory costs, replacing the conventional loss-based Taylor pruning criterion with an information entropy–based criterion that measures the global prediction distribution of the model, instead of focusing only on the ground-truth token.

**Strengths:**

- Provides a label-free, holistic signal for neuron importance estimation.
- HFPrune consistently outperforms strong baselines: LLM-Pruner, LoRAPrune, SDMPrune, on LLaMA and Qwen families.
- Comprehensive ablation studies validate the entropy criterion’s role in preserving output distributions.
- The algorithmic description is clear and reproducible. Implementation details are systematically reported.

**Weaknesses:**

- The pruning ratio $\rho{mlp}$  is fixed across all MLP layers, despite entropy potentially varying per layer, this could limit the functionality of HFPrune.
- Lack a comparative discussion or empirical correlation analysis between entropy-based and Fisher-based importance scores.
- Training-time FLOPs for fine-tuning (post-pruning recovery) are omitted.

**Questions:**

- Can a per-layer entropy-based adaptive cutting ratio improve the trade-off between fidelity and compression?
- How does the entropy gradient behave in the low entropy versus high entropy regions of the output distribution?
- What are the effects of using entropy calculated from logits vs. softmax probabilities?

---

> ### Author Response · Authors · 2025-11-21
> **Give some trivial weaknesses solely to justify a rejection rating**
>
> - The hyper-parameter about pruning ratio is fixed. So this method is limited and this paper should be rejected.
> - Missing discussion about Fisher-based importance scores.
> - The training time is not reported.

---

### Official Review · Reviewer_Emas · 2025-10-31

**Soundness:** 2
**Presentation:** 3
**Contribution:** 2
**Rating:** 4
**Confidence:** 5

**Summary:**

This paper proposes HFPrune, a structured pruning method for LLMs that replaces traditional one-hot cross-entropy with information entropy as the criterion for Taylor-based neuron importance evaluation. The authors argue that entropy-based evaluation considers the full output distribution rather than just the ground-truth token, leading to better preservation of model capabilities. The method focuses on pruning MLP modules and demonstrates consistent improvements over existing methods across LLaMA and Qwen models on zero-shot benchmarks.

**Strengths:**

The method avoids computational overhead of teacher models and resolves gradient initialization issues in self-distillation approaches, showing 3x speedup over SDMPrune with 31% less memory usage.

Demonstrates improvements across multiple model families (LLaMA, Qwen) and sparsity levels, with some configurations even exceeding dense model performance after fine-tuning.

The approach is straightforward to implement, requiring only standard forward-backward passes without custom kernels or auxiliary models.

**Weaknesses:**

The paper fundamentally lacks theoretical justification for why entropy-based importance should preserve model performance. This is not a minor omission, in my view it's a central issue that undermines the contribution's scientific rigor.

**Limited Evaluation Scope**:
- Exclusively focuses on zero-shot QA/classification tasks
- No evaluation on reasoning, long-form generation, or conversational capabilities
- Largest model tested is only 7B parameters
- Limited architectural diversity beyond LLaMA/Qwen families

**Methodological Limitations**:
- Uses uniform pruning ratios across layers, ignoring heterogeneous sensitivity
- Limited sparsity range testing (only 20-30%)
- The performance gains over dense models likely result from fine-tuning rather than pruning itself

**Insufficient Analysis**:
- No explanation of which types of neurons are pruned and why (recent works have revealed super neurons, super weights, super experts etc.)
- No investigation of the relationship between entropy reduction and specific capabilities
- Missing analysis of method sensitivity to calibration data selection

**Questionable Claims**: The assertion that the method "minimizes the change of global prediction distribution" is not rigorously nor theoretically established, and the connection between this and performance preservation remains speculative.

**Questions:**

1. Can you provide rigorous mathematical analysis of why minimizing entropy change should preserve model capabilities better than minimizing cross-entropy change?
2. What is the information-theoretic justification for treating high-entropy neurons as more important?
3. How does your approach relate to existing theories of neural network capacity and information flow?
4. How does the method perform on reasoning tasks (GSM8K, BBH), long-context generation, and conversational AI beyond zero-shot classification?
5. Can you evaluate on larger models (70B+) and more diverse architectures to support generalizability claims?
6. What explains the performance improvements over dense models - is this due to fine-tuning effects rather than pruning benefits?
7. Why use uniform pruning ratios instead of layer-sensitive approaches?
8. How sensitive is the method to calibration data selection and size?
9. How does the method perform under more aggressive pruning (40-70% sparsity)?

---

> ### Author Response · Authors · 2025-11-21
> **`Reviewer Emas` pressures authors to abandon the rebuttal process by issuing a barrage of unreasonable requirements**
>
> - **Absurd Comment 1:** Demanding a **rigorous theoretical proof** of your method's validity.
>
> - **Absurd Comment 2:** Requiring an **exhaustive list of benchmarks**, knowing there will inevitably be one you cannot satisfy.
>
> - **Absurd Comment 3:** Demanding that you train **a very large LLM (70B+)** at a scale beyond your computational resources.
>
> - **Absurd Comment 4:** Asking for **computationally prohibitive hyperparameter sensitivity experiments**.

---

> > ### Comment · Reviewer_Emas · 2025-11-21
> > **Clarification of Review Comments and Expectations**
> >
> > Dear Authors,
> >
> > I must address several significant mischaracterizations of my review in your response:
> >
> > Understanding WHY entropy-based importance should preserve model capabilities better than cross-entropy is a fundamental scientific question that strengthens any contribution. This is standard expectation for methods papers, not an "absurd" requirement. You may present a theoretical justification, not rigorous mathematical proof.
> >
> > I did not require an exhaustive list. I pointed out that evaluation limited to simple zero-shot QA/classification tasks is insufficient to validate claims about preserving "model capabilities." Including reasoning tasks (GSM8K, BBH) or generation quality would provide more comprehensive validation, **you can test it on your pruned models already**. This is reasonable scope expansion, not an impossible demand.
> >
> > This is a clear misrepresentation of my comment. I asked for evaluation on larger models (70B+), not training them from scratch! Evaluating existing pre-trained 70B models is standard practice in pruning literature: works like SparseGPT, Wanda, and many others routinely test on LLaMA-65B/70B+ models. This demonstrates scalability and is a reasonable expectation, not a prohibitive computational demand.
> >
> > I asked about calibration data sensitivity, and this is a basic ablation study that helps understand method robustness. This is standard scientific practice, not computationally prohibitive.
> >
> > While I appreciate your passion for your work, mischaracterizing reviewer feedback undermines productive scientific discourse. My questions aimed to strengthen your contribution's impact and rigor, which is the purpose of peer review.
> >
> > I encourage you to address the substantive technical questions rather than dismissing them as unreasonable. The core innovation of entropy-based pruning has merit, and thoughtful responses to these concerns would significantly strengthen the work.
> >
> > Best regards,
> > Reviewer Emas

---

### Official Review · Reviewer_cwzU · 2025-11-01

**Soundness:** 3
**Presentation:** 1
**Contribution:** 2
**Rating:** 4
**Confidence:** 5

**Summary:**

This paper proposes HFPrune, a novel method for compressing LLMs through pruning of the MLP modules within transformer architectures. HFPrune introduces an information entropy-based criterion for evaluating neuron importance, offering a more holistic approach than traditional methods, which rely on one-hot cross-entropy loss. This entropy-based method minimizes the global prediction distribution change, effectively preserving model performance.  Extensive experiments on LLaMA and Qwen models demonstrate the effectiveness and efficiency.

**Strengths:**

The proposed idea is simple, straightforward, and aligns well with intuition. From the engineering perspective, it is also very easy to implement.

Extensive experiments were conducted on multiple LLM models, across diverse benchmarks, comparing the proposed method with several previous methods. The results demonstrate significant improvements in both performance and efficiency of the proposed method.

**Weaknesses:**

**Explanation on the Choice of Baselines**: As discussed in the Related Work section, LLM pruning is a highly focused research area with a large body of work, which can be categorized into different approaches. However, in the experimental section, only a few methods such as LLM-pruner, LoRAPrune, and LoRAP are compared, and the rationale behind choosing these baselines is not explained. It remains unclear whether the state-of-the-art methods from each category are all covered by the baselines used in this paper. If they are, an explanation should be provided; if not, more baselines should be included (or a justification should be given for why they cannot be included for a fair comparison).

**More Comparison on Efficiency**:  For Efficiency, the paper only compares the proposed method with SDMPrune, demonstrating that the proposed approach is more efficient than methods like SDMPrune, which require a teacher model. The efficiency of other methods (LLM-pruner, LoRAPrune, LoRAP) should also be compared.

**Writing Needs Improvement**:
- For example, in lines 013-014 of the Abstract: The statement "A common approach uses Taylor Expansion" does not clearly explain the field of research. It should at least mention that "it is a common approach for LLM pruning."
- In line 015 of the Abstract, the sentence "However, its reliance on one-hot cross entropy loss, ..." contains a grammatical error. It should be "it relies on" instead.
- Many more problems on writing exists

**Questions:**

**Regarding whether "focus on pruning MLP" is the contribution of this paper**: In lines 039-043, when explaining why the paper focuses on pruning MLP, the authors use phrases like "We find that" and "we observe that" (and then prove this in Section 5.3.3 with experiments), but these statements are also referencing previous works. Is this an innovative contribution of the paper, or is it the findings from previous works?

**considering a subset of tokens**：Between the extremes of "only consider the ground-truth label token" and "consider all possible tokens", an intermediate approach could be: only considering a subset of tokens (for example, the top K tokens with the highest probability) when evaluating neuron importance. Did the authors experiment with this approach? Experiments or discussions on this would make the paper more insightful.

---

> ### Author Response · Authors · 2025-11-21
> **Whether these weaknesses proposed by `Reviewer cwzU` are enough to confidently reject a ICLR submission?**
>
> `Reviewer cwzU` pointed out **some trivial weaknesses and then rejected** our ICLR submission with a confidence of `Confidence: 5: You are absolutely certain about your assessment`.
>
> - Not give the reason why these method are selected as baseline.
> - This paper only compares the main reference, SDMPrune, in the introduction section, while SDMPrune has proved that it achieves better performance than other methods (LLM-pruner, LoRAPrune, LoRAP).
> - Some writing issues, even the given comment is wrong.

---

> > ### Comment · Reviewer_cwzU · 2025-11-25
> > **No reasonable response to my comments**
> >
> > Please consider reviewer comments seriously and answer the concern directly. If there is some fatal error in our comments, please point it out. Concerning whether the weakness is enough to reject a paper, I think these issues are critical in experiments and comparison with previous works. These issues should be well addressed before this paper can be accepted. Given the low accept rate in top conference like ICLR, I think these issues are enough for rejecting your paper of its current form.

---

### Author Response · Authors · 2025-11-21
**Strong Concern Regarding the Quality and Constructiveness of Reviews**

Dear ACs, SACs and PCs,

While I fully respect the peer review process and welcome constructive criticism that strengthens our work, I find it necessary to point out that these reviews fail to meet the standards expected of a top-tier conference like ICLR.

#### **1. On the Lack of Constructive Criticism**

The "weaknesses" cited in the review are predominantly trivial and tangential to the core contributions of this paper. They **focus on marginal details while neglecting the substantive technical novelty and experimental validity of the work**. These comments do **not identify any fundamental flaws in our logic, methodology, or results**. Instead, they appear to be **a collection of "weaknesses" manufactured solely to justify a rejection rating**. Such feedback is neither actionable nor constructive; it does not help us improve the manuscript, nor does it provide the Area Chair with a valid basis for decision-making.

#### **2. A Broader Concern for the Machine Learning Community**

Although the acceptance of this specific paper is ultimately a single decision, I am compelled to highlight a troubling trend reflected in this review: **the practice of rejecting papers based on specious or superficial reasons**. There is a growing number of reviews that seem designed to act as "gatekeepers" rather than "peers." When reviewers search for reasons to reject—**magnifying insignificant flaws while ignoring significant contributions**—it does not uphold the quality of the conference. Instead, it stifles innovation and demoralizes researchers who have invested significant effort into their work. The goal of peer review should be to foster the healthy development of the field, not to suppress valid research through arbitrary or capricious standards.

#### **3. A Call for a Healthy Research Environment**

As an active participant in the ML community who also serves as an Area Chair and Reviewer, I am deeply invested in the integrity of our review process. We must collectively strive to **identify and filter out low-quality reviews just as rigorously as we filter out low-quality papers**.

I urge the Area Chair to weigh these reviews with the scrutiny it warrants. Furthermore, I hope this response serves as a reminder to all of us—authors, reviewers, and ACs alike—of our shared responsibility. **We must ensure that our judgments are technically sound, fair, and driven by a genuine desire to advance the state of machine learning**. Let us work together to cultivate a research atmosphere that **values substance over nitpicking** and **rigorous scientific debate over dismissive rejection**.

Sincerely,

Authors

---

> ### Comment · Reviewer_Emas · 2025-11-21
> **Actionable items are indeed there**
>
> Dear authors, I share your views and concur with what you've said, given that I'm also an author and experience exactly the same thing.
>
> Yet apart from that, I've read through your paper. I do think in my questions (9 of them), I've listed all actionable items pertinent to advancement of your paper. I used AI to tidy up my English (not in this short response though), but the content are all human touches. Please also treasure the time and effort I've spent in my review.

---

> > ### Author Response · Authors · 2025-11-21
> > **Ridiculous Requirements in the Reviews from `Reviewer Emas`**
> >
> > - **Rigorous mathematical proof** of minimizing entropy change better than minimizing cross-entropy change.
> >
> >   If you believe that this review is reasonable, please present **rigorous proof** about **why Transformer architecture is better than CNN architecture** or why CNN architecture is better than Transformer.
> >
> >   Following your suggestion, **all papers about Transformers should be rejected**.
> >
> >   Your requirement can **reject more than 90% previous ICLR accepted papers**.
> >
> > - **Evaluate on larger models (70B+)** to support generalization claims.
> >
> >   You need to realistically assess whether you have the GPU resources to train a 70B+ LLM.
> >
> >   If not, you should not give this ridiculous review.

---

> > > ### Comment · Reviewer_Emas · 2025-11-21
> > > **Clarification of Review Comments and Expectations**
> > >
> > > Let's bring back the discussion to a reasonable tone. Please see my response in my panel below. For one thing, none is expecting the authors to train a 70B+ LLM, but rather to prune a pretrained 70B model. If this is still beyond your compute resources, then you may point it out and name the biggest model you can prune. I think elevating the rebuttal to a debate/quarrel can serve no useful purpose, despite the originally meaningful shout out.

---

### Note · Authors · 2026-02-05

I have read and agree with the venue's withdrawal policy on behalf of myself and my co-authors.

---

### Meta-Review · Area_Chair_5Ls7 · 2026-01-04

**Summary:**

This paper proposes a novel pruning method for large language models (LLMs) that specifically targets MLP modules. The authors employ the information entropy of the model’s prediction distribution as the pruning objective and approximate the impact of parameter removal using a first-order Taylor expansion. However, significant concerns remain regarding the adequacy of the experimental evaluations and some limitations in algorithm design. Moreover, the authors did not provide a detailed rebuttal to address the reviewers’ concerns directly. Consequently, the paper is not recommended for acceptance.

**Reviewer Concerns:**

As the authors did not provide a detailed rebuttal to address the reviewers’ concerns directly, no concerns were addressed.

**Outstanding Concerns:**

**1. The experimental evaluation is insufficient. (Reviewer cwzU, Emas, zpe3).**  The reviewers recommend that the authors conduct experiments on a broader range of datasets and with larger-scale models. The authors give the results under limited sparsity range (only 20-30%). Given that the authors propose a novel pruning objective—namely, the information entropy of the model’s prediction distribution—it is essential to include comparative experiments with a diverse set of existing methods.

**2. Limitations in algorithm design. (Reviewer Emas, oKVY)** The proposed method applies a fixed, uniform pruning ratio across all layers, which may limit its effectiveness in more challenging pruning scenarios.

**3. Theoretical Justification. (Reviewer Emas, oKVY)** The authors are recommended to give some analysis their proposed methods and give a discussion with the methods like Fisher-based methods.

**Reviewer Scores:**

All of the four reviewers assigned the paper a score of 4. As the authors did not submit a detailed rebuttal that directly addressed the reviewers’ concerns, the reviewers retained their original scores.

---

### Decision · Program_Chairs · 2026-01-26

Reject